# The Sigma-1 Receptor Exacerbates Cardiac Dysfunction Induced by Obstructive Nephropathy: A Role for Sexual Dimorphism

**DOI:** 10.3390/biomedicines12081908

**Published:** 2024-08-20

**Authors:** Francisco Javier Munguia-Galaviz, Alejandra Guillermina Miranda-Diaz, Yanet Karina Gutierrez-Mercado, Marco Ku-Centurion, Ricardo Arturo Gonzalez-Gonzalez, Eliseo Portilla-de Buen, Raquel Echavarria

**Affiliations:** 1Departamento de Fisiologia, Centro Universitario de Ciencias de la Salud, Universidad de Guadalajara, Guadalajara 44340, Jalisco, Mexico; francisco.munguia4072@alumnos.udg.mx (F.J.M.-G.); alejandra.miranda@academicos.udg.mx (A.G.M.-D.); 2Division de Ciencias de la Salud, Centro Universitario del Sur, Universidad de Guadalajara, Ciudad Guzman 49000, Jalisco, Mexico; 3Departamento de Clinicas, Centro Universitario de los Altos, Universidad de Guadalajara, Tepatitlan 47620, Jalisco, Mexico; yanet.gutierrez@academicos.udg.mx; 4Unidad de Biotecnologia Medica y Farmaceutica, Centro de Investigación y Asistencia en Tecnología y Diseño del Estado de Jalisco (CIATEJ), Guadalajara 44270, Jalisco, Mexico; maku_al@ciatej.edu.mx; 5Division de Investigacion Quirurgica, Centro de Investigacion Biomedica de Occidente, Instituto Mexicano del Seguro Social, Guadalajara 44340, Jalisco, Mexico; qfbragg@gmail.com (R.A.G.-G.); eliseo.portilla@live.com.mx (E.P.-d.B.); 6Consejo Nacional de Humanidades, Ciencias y Tecnologias (CONAHCYT)—Centro de Investigacion Biomedica de Occidente, Instituto Mexicano del Seguro Social, Guadalajara 44340, Jalisco, Mexico

**Keywords:** Sigmar1, kidneys, cardiorenal, fibrosis, sexual dimorphism

## Abstract

The Sigma-1 Receptor (Sigmar1) is a stress-activated chaperone and a promising target for pharmacological modulation due to its ability to induce multiple cellular responses. Yet, it is unknown how Sigmar1 is involved in cardiorenal syndrome type 4 (CRS4) in which renal damage results in cardiac dysfunction. This study explored the role of Sigmar1 and its ligands in a CRS4 model induced by unilateral ureteral obstruction (UUO) in male and female C57BL/6 mice. We evaluated renal and cardiac dysfunction markers, Sigmar1 expression, and cardiac remodeling through time (7, 12, and 21 days) and after chronically administering the Sigmar1 agonists PRE-084 (1 mg/kg/day) and SA4503 (1 mg/kg/day), and the antagonist haloperidol (2 mg/kg/day), for 21 days after UUO using colorimetric analysis, RT-qPCR, histology, immunohistochemistry, enzyme-linked immunosorbent assay, RNA-seq, and bioinformatics. We found that obstructive nephropathy induces Sigmar1 expression in the kidneys and heart, and that Sigmar1 stimulation with its agonists PRE-084 and SA4503 aggravates cardiac dysfunction and remodeling in both sexes. Still, their effects are significantly more potent in males. Our findings reveal essential differences associated with sex in the development of CRS4 and should be considered when contemplating Sigmar1 as a pharmacological target.

## 1. Introduction

Patients suffering from chronic kidney disease (CKD) present a higher risk of cardio-vascular complications, contributing to approximately 50% of all deaths in advanced-stage CKD patients [1]. Cardiorenal syndrome type 4 (CRS4) is the pathology in which acute or chronic renal damage gives rise to cardiac dysfunction [2]. Although the initial activation of adaptive mechanisms aims to maintain normal renal function and cardiac output, with time it becomes maladaptive, causing a progressive decline in cardiac performance mainly due to myocardial and vascular remodeling. Renal dysfunction transforms the interactions between hemodynamic, neurohormonal, uremic, immunological, and inflammatory factors that maintain the physiological homeostasis of the kidneys and heart, initiating a multiorgan dysfunction [3]. Furthermore, alterations in the tissue microenvironment of both organs force the cells to activate stress-associated pathways involved in inflammation and fibrogenesis [4]. 

The Sigma-1 Receptor (Sigmar1) is a stress-activated chaperone protein mainly residing in the mitochondria-associated endoplasmic reticulum (ER) membrane domain (MAM) that can translocate to the cell membrane, mitochondria, and nucleus upon stimulation [5,6,7]. Sigmar1 influences cell-signaling pathways by interacting with various ligands and proteins, and its ability to induce specific cellular responses makes it a promising target for pharmacological modulation [8]. Sigmar1 agonists, such as PRE-084 and SA4503, and antagonists like haloperidol, promote opposing, ligand-mediated conformational changes that result in distinct oligomeric states influencing Sigmar1’s ability to form a complex with the ER chaperone BiP and thus control the function of downstream signaling molecules [9,10,11]. The kidney and heart express Sigmar1, and accumulating data emphasize Sigmar1’s functions in organ protection, ischemic injury, myofibroblast activation, and fibrosis [12]. Yet, it is unknown how Sigmar1 is involved in CRS4 and whether Sigmar1 agonists and antagonists influence renal and cardiac dysfunction.

The existence of sex-dependent disparities in kidney and cardiovascular diseases is well documented, and the cardiorenal phenotypes of men and women are markedly different [13,14]. Hypertension, atrial fibrillation, hyperkalemia, and heart failure with reduced ejection fraction are more frequent in male patients with CRS4. In contrast, female patients more often exhibit congestion and heart failure with preserved ejection fraction. However, our insight into the molecular mechanisms responsible for the differences in diagnosis, presentation, progression, and treatment response associated with sex is limited [15]. Evidence suggests that sex and hormones affect Sigmar1 function. Hosszu A et al. demonstrated the sexual dimorphism of Sigmar1 in a murine renal ischemia-reperfusion (IR) injury model [16]. Sigmar1 activation by 17-beta-estradiol enhances the heat shock response in the kidneys, thus conferring renoprotection. Similarly, Wilbert-Lampen U et al. found that Sigmar1 mediates estradiol- and progesterone-dependent endothelin 1 release in endothelial cells, but not testosterone, and proposed that male and female sex hormones can bind to Sigmar1 at different sites to perform antagonistic effects [17]. To date, how Sigmar1 operates in the context of CKD-induced cardiac injury and whether sex plays a part remains to be clarified. 

Previously, we demonstrated that unilateral ureteral obstruction (UUO) is a moderately uremic model resembling early CKD able to induce cardiac transcriptomic changes implicated in cellular pathways associated with cardiovascular disease development [18]. Given the importance of sexual dimorphism and stress-activated signaling pathways, this study aimed to explore the function of Sigmar1, its agonists (PRE-084 and SA4503), and antagonist (haloperidol) in a CRS4 model induced by UUO in male and female mice. We found that obstructive nephropathy induces Sigmar1 expression in the kidneys and heart, and that Sigmar1 stimulation with its agonists aggravates cardiac dysfunction and remodeling in both sexes but significantly more in males.

## 2. Materials and Methods

### 2.1. Study Animals

We used 12–16-week-old male and female C57BL/6 mice bred, housed, and maintained in our research animal facility following institutional and federal regulations (NOM-062-200-1999), the Animal Research: Reporting of In Vivo Experiments (ARRIVE) guidelines, and the Guide for the Care and Use of Laboratory Animals from the National Research Council. This study received approval from the Local Health Research Committee of CIBO, IMSS (CLIS R-2021-1305-013).

### 2.2. UUO Surgical Model and Pharmacological Administration of Sigmar1 Agonists and Antagonist 

The murine UUO surgery was carried out according to Hesketh E et al. [19]. We performed a midline celiotomy in mice under xylazine (10 mg/kg) and ketamine (100 mg/kg) i.m. anesthesia, carefully monitoring physiological parameters such as heart rate, respiratory rate, and deep pain reflexes, and supplementing isoflurane through open-circuit inhalation when further analgesic deepening was needed. In male and female mice subjected to UUO surgery, we exposed, ligated, and cut the left ureter before closing the muscular plane with 3-0-gauge silk sutures and the cutaneous plane with 5-0-gauge nylon sutures. In the case of sham surgery, the muscular and skin planes were immediately closed after the abdominal incision without ligation. We maintained the animals in a temperature-controlled incubator until they recovered from the anesthesia and the surgical procedures. Then, we placed them in clean cages, with water and food *ad libitum*, for specified times (7, 12, or 21 days). Once the post-surgery time of the different groups had expired, we administered xylazine (10 mg/kg) and ketamine (100 mg/kg) i.m. anesthesia and collected cardiac blood, the left kidney, and the heart. The blood was processed to obtain serum by centrifuging at 3500× *g* for 15 min at 4 °C. After dissection, the organs were frozen at −80 °C for RNA isolation and protein extraction, or placed in 10% formalin in PBS pH 7.4 for 24 h before paraffin embedding for histological analysis (Figure 1A). To determine the effect of Sigmar1 agonists and antagonists in our CRS4 model, we performed the UUO surgical procedure as previously described, with the difference that control vehicle (saline, NaCl 0.9%), Sigmar1 agonists PRE-084 (1 mg/kg) and SA4503 (1 mg/kg), and the Sigmar1 antagonist haloperidol (2 mg/kg) were administered i.p. daily, starting at 24 h post-surgery, until day 21 when we euthanized the animals and processed their blood and tissues, whole heart, and kidney for the subsequent analysis of renal function, cardiac hypertrophy markers, and Sigmar1 (Figure 1B). 

### 2.3. Colorimetric Tests for Renal Function 

To determine serum creatinine (sCR) and blood urea nitrogen (BUN) levels, we used colorimetric assays (Valtek Diagnostics, Santiago, Chile) as instructed by the manufacturer. We recorded the absorbances using an EPOCH spectrophotometer (BioTek, Winooski, VT, USA).

### 2.4. RNA Extraction and RT-qPCR

To extract total RNA from kidney and heart tissues, we used Trizol (Invitrogen, Carlsbad, CA, USA) and, afterward, synthesized cDNA with the RevertAid First Strand cDNA Synthesis Kit (Thermo Fisher Scientific, Waltham, MA, USA). Then, we performed RT-qPCR in a QuantStudio™ 5 instrument (Thermo Fisher Scientific, Waltham, MA, USA) with Green-2-Go-qPCR-Mastermix-ROX (BioBasic, Markham, ON, Canada) using the specific primer pairs listed in Appendix A. We applied the delta-CT method to obtain differences in gene expression, expressed as fold-change, after normalizing with the glyceraldehyde 3-phosphate dehydrogenase (Gapdh) gene.

### 2.5. Histology and Immunohistochemistry (IHC)

Paraffin-embedded tissues sectioned into 5-μm were dewaxed, rehydrated, and treated with the histological stain Sirius Red. Additionally, we conducted immunohistochemistry on these sections. We used a citrate buffer (10 mM, pH 6.0) at 95–100 °C for 3 min to recover the antigens, coupled with the peroxidase-based Novolink Polymer Detection System (Leica Biosystems, Wetzlar, Germany) and polyclonal primary antibodies against Sigmar1 (ABCAM, Cambridge, UK), 1:100 dilution, and smooth muscle actin alpha 2 (Acta2) (CST, Danvers, MA, USA), 1:200 dilution. We recorded the tissue images in a DMi1 light microscope at 200× (Leica, Wetzlar, Germany), analyzed them using the NIH ImageJ/Fiji software 1.54f, and calculated the percentage (%) positive area of Sirius Red, Sigmar1, and Acta2 using the following formula: % Area = 100 × (Area of interest/Total area) in three fields per animal.

### 2.6. Enzyme-Linked Immunosorbent Assay (ELISA)

To determine the concentrations of fibroblast growth factor 23 (FGF-23) and transforming growth factor beta (TGF-β) in tissue and serum samples, respectively, we used specific DuoSet ELISA kits (R&D Systems, Minneapolis, MN, USA) following the manufacturer’s instructions. We normalized the FGF-23 tissue concentration to the total protein concentration in the lysate, which was quantified using the Bradford protein assay (ABP Biosciences, Beltsville, MD, USA). We measured the absorbances in an EPOCH spectrophotometer (BioTek, Winooski, VT, USA).

### 2.7. RNA Sequencing (RNA-Seq) and Bioinformatic Analysis

To understand how Sigmar1 activation with PRE-084 affects the mRNA expression profiles in the hearts of male mice 21 days after UUO surgery, we performed RNA-seq (N = 3). We first prepared a poly(A) RNA-seq library following the sample preparation protocol of Illumina’s TruSeq stranded mRNA (Illumina, San Diego, CA, USA) in RNA samples that passed all integrity checks (RIN > 7). Poly (A) tail-containing mRNAs were purified using oligo-(dT) magnetic beads followed by fragmentation at high temperatures. We used the Agilent Technologies 2100 Bioanalyzer High Sensitivity DNA Chip for quality control and quantification of the sequencing library. We performed paired-end sequencing on Illumina’s NovaSeq 6000 system (LC Sciences, Houston, TX, USA), and we submitted the raw sequence data to the NCBI Gene Expression Omnibus (GEO) repository under accession code GSE235751.

Before transcript assembly, we removed low-quality reads with Cutadapt v1.10 and in-house perl. Then, we verified sequence quality with FastQC 0.10.1 (http://www.bioinformatics.babraham.ac.uk/projects/fastqc/, accessed on 1 March 2023), and HISAT2 2.0 was used to map reads to ftp://ftp.ensembl.org/pub/release-101/fasta/mus_musculus/dna/genome (https://hpc.nih.gov/refdb/dbview.php?id=646), accessed on 1 March 2023. We assembled the mapped reads of each sample using StringTie, and we achieved the reconstruction of a complete transcriptome by merging all transcriptomes using perl scripts and gffcompare (https://github.com/gpertea/gffcompare, accessed on 1 March 2023). For the differential expression analysis of mRNAs, we used StringTie 1.3.4 and ballgown (http://www.bioconductor.org/packages/release/bioc/html/ballgown.html, accessed on 1 March 2023), calculating Fragments per Kilobase per Million (FPKM) to estimate expression levels, and mRNA differential expression analysis using the R package DESeq2. mRNAs with a false discovery rate (FDR) < 0.05 and absolute fold-change ≥ 2 were considered differentially expressed genes (DEGs). Enrichment analysis for DEGs was performed using the Gene Ontology (GO) project (http://www.geneontology.org, accessed on 1 March 2023) and enrichment analysis for functional interpretation of the RNA-seq experimental data.

### 2.8. Statistical Analysis

We used the GraphPad Prism 5.0 software (GraphPad Software, Inc., San Diego, CA, USA) for data analysis, presenting quantitative data as mean ± standard error of the mean (SEM). We used the one-way ANOVA (parametric) or Kruskal–Wallis (non-parametric) tests to determine statistically significant differences between the means of three or more independent samples as appropriate, considering any value of *p* < 0.05 to be statistically significant.

## 3. Results

### 3.1. UUO Effectively Induces CRS4 Development 

We performed UUO to evaluate renal and cardiac changes induced at 7, 12, and 21 days post-surgery in male and female mice. UUO augmented the size and weight of the ligated left kidney, accompanied by morphological changes such as fluid accumulation and loss of parenchyma (Figure 2A). When we evaluated markers of kidney damage, we found no differences in sCr levels between groups, except for females after 21 days of UUO when sCr was significantly increased (Figure 2B). In contrast, BUN levels progressively accumulated after UUO, reaching a substantial increment at 12 days post-surgery in both males (42.10 ± 2.43 mg/dL; *p* < 0.05) and females (43.01 ± 7.55 mg/dL; *p* < 0.05), and returning to sham levels by day 21 (Figure 2C). Interestingly, UUO significantly upregulated neutrophil gelatinase-associated lipocalin (Ngal) gene expression in a time-dependent manner, but only in the left kidneys of females at days 7 (19.56 ± 3.60-fold; *p* < 0.05) and 12 (40.86 ± 21.88-fold; *p* < 0.05), remaining high after 21 days (Figure 2D).

To evaluate the development of de novo heart disease after renal injury, we determined expression levels of atrial natriuretic peptide (Anp), brain natriuretic peptide (Bnp), and smooth muscle actin alpha 2 (Acta2), which are sensitive markers of hemodynamic burden and cardiac remodeling [20]. Cardiac Anp expression increased at day 7 post-UUO in both males (8.53 ± 1.13-fold; *p* < 0.01) and females (8.76 ± 2.62-fold; *p* < 0.05), returning to lower levels after 12 days (Figure 2E). UUO did not induce Bnp cardiac expression (Figure 2F), but after 21 days, UUO significantly increased Acta2 gene expression in male hearts (40.51 ± 28.71-fold; *p* < 0.05) but not in females (Figure 2G). Still, when we assessed collagen deposition in cardiac tissues through Sirius Red staining, UUO significantly augmented the percentage of collagen-positive areas in both male (6.95 ± 0.85% Sirius red-positive area; *p* < 0.001) and female (7.21 ± 0.89% Sirius red-positive area; *p* < 0.001) mice after 21 days (Figure 2H,I). In conclusion, the UUO model effectively induces the development of CRS4 in a time- and sex-dependent manner.

### 3.2. Obstructive Nephropathy Elevates Sigmar1 Expression in the Kidneys and Heart 

Since Sigmar1 participates in cellular signaling under stress conditions [6], we investigated the effect of UUO on Sigmar1 expression in the kidneys and the heart. Figure 3A shows representative images of Sigmar1 immunohistochemical staining in kidney tissues of male and female mice 7, 12, and 21 days after UUO surgery. Sigmar1-positive areas mainly coincided with tubular structures, though some Sigmar1 staining also appeared in the glomerulus. Sigmar1 protein labeling intensity increased with UUO progression, and the percentage of the Sigmar1-positive area was significantly higher compared to sham in males at 12 (47.41 ± 3.88; *p* < 0.05) and 21 days (64.63 ± 6.05; *p* < 0.001), and females at 12 (43.46 ± 5.09; *p* < 0.05) and 21 days (66.21 ± 4.35; *p* < 0.001) (Figure 3B). UUO also augmented Sigmar1 mRNA expression at least three-fold in both male and female kidneys (Figure 3C).

When we analyzed the cardiac tissues of male and female mice, we found intense labeling of the Sigmar1 protein in both sham and UUO (Figure 3D). Although Sigmar1-positive areas tended to be larger at 12 days post-surgery, Sigmar1 quantification in these tissues did not show statistical differences between groups (Figure 3E). Nevertheless, Sigmar1 mRNA expression in cardiac tissues augmented as the renal lesion advanced, being highest and statistically significant 12 days after UUO in males (34.01 ± 9.67-fold; *p* < 0.05) and females (7.52 ± 2.90-fold; *p* < 0.05); returning to sham levels by day 21 (Figure 3F). Notably, UUO-induced Sigmar1 mRNA expression, but not protein, was much higher in male hearts.

### 3.3. Male Sex Potentiates Sigmar1 Effects on Renal and Cardiac Function Markers after UUO 

To explore whether activating or inhibiting Sigmar1 after UUO affects renal and cardiac injury markers, we administered a daily i.p. dose of Sigmar1 agonists PRE-084 (1 mg/kg/day) or SA4503 (1 mg/kg/day) and Sigmar1 antagonist haloperidol (2 mg/kg/day) (Figure 1B). Both agonists decreased sCr levels, with a statistically significant reduction (*p* < 0.05) of > 50% in males and females by PRE-084 and SA4503, compared to animals receiving saline (Figure 4A). PRE-084 administration also decreased BUN levels by 11.40% in males (*p* < 0.05) (Figure 4B). In contrast, both Sigmar1 agonists and haloperidol increased Ngal gene expression by at least two-fold (PRE-084 and haloperidol; *p* < 0.05), but only in male kidneys (Figure 4C).

Regarding cardiac injury markers, Sigmar1 agonists and haloperidol augmented Anp and Bnp gene expression after UUO (Figure 4D,E). However, Anp and Bnp induction by Sigmar1 agonists was higher, more than 100-fold, and statistically significant (*p* < 0.05) only in male hearts. Additionally, we quantified renal FGF-23 due to its importance in mediating kidney–heart communication. Still, we did not observe significant changes in FGF-23 renal concentration in response to treatment with Sigmar1 agonists or haloperidol (Figure 4F).

### 3.4. Sigmar1 Agonists and Antagonists Potentiate UUO-Induced Cardiac Fibrosis Regardless of Sex 

Cardiac fibrosis is a primary pathological process in the progression of CRS4, causing structural changes responsible for mechanical function anomalies and elevated oxygen needs [21]. Since UUO enhances cardiac fibrosis 21 days after surgically provoking the renal lesion, we determined the effects of Sigmar1 agonism and antagonism on this process using Sirius Red staining. Both agonists (PRE-084 and SA4503) increased cardiac fibrosis (*p* < 0.05) compared to saline-receiving mice, observed as almost double the percentage of Sirius Red-positive areas (Figure 5A,B). In contrast, only haloperidol was able to increase mRNA expression of collagen type I alpha 1 chain (Col1a1), being only statistically significant in female hearts (8.85 ± 3.77-fold; *p* < 0.05), but this increase failed to translate into protein (Figure 5B,C).

Figure 5D shows representative images of Acta2 staining in cardiac tissues of male and female mice 21 days after UUO surgery that received saline, PRE-084, SA4503, or haloperidol. We observed that Acta2-positive areas increased in the hearts of animals treated with Sigmar1 agonists or antagonist, regardless of sex. The Acta2-positive area was more than 60% and 80% in animals chronically receiving Sigmar1 agonists or antagonist compared to 25.24% ± 4.84 and 61.26% ± 5.31 in saline-treated males and females, respectively (Figure 5E). In contrast, all Sigmar1 ligands significantly decreased Acta2 mRNA expression in males (*p* < 0.05) (Figure 5F). Finally, we determined serum levels of TGF-β, a key fibrosis mediator [22]. Though a slight decrease in TGF-β-circulating levels was observed in haloperidol-treated females 21 days after UUO, the effect of none of the treatments was statistically significant (Figure 5G).

### 3.5. Sigmar1 Expression in the Kidneys and Heart Is Modulated by Its Agonists and Antagonists after UUO 

Sigmar1 ligands influence its expression in various cells and tissues, and elevated Sigmar1 levels could be partly responsible for the changes observed in the kidneys and hearts of UUO mice associated with CRS4 development [23,24,25]. Hence, we examined how Sigmar1 expression was affected by the chronic administration of its agonists and antagonist. Kidneys of UUO mice showed some differences in Sigmar1-positive areas in IHC staining (Figure 6A,B), particularly in response to PRE-084, but they were not statistically significant. However, elevated Sigmar1 gene expression induced by SA4503 (2.22-fold ± 0.31) and haloperidol (3.12-fold ± 0.70) was present in the kidneys of females (*p* < 0.05) (Figure 6C). In the heart, only PRE-084 reduced Sigmar1-positive areas, 65.05% ± 5.74 vs. saline, 80.41% ± 10.75 (*p* < 0.05) in male mice (Figure 6D,E). Meanwhile, SA4503 was the only pharmacological treatment that increased cardiac Sigmar1 gene expression compared to saline UUO by 2.84-fold ± 0.74 in females (Figure 6F).

### 3.6. Sigmar1 Agonist PRE-084 Induces Transcriptomic Changes Associated with Pathological Cardiac Remodeling

Since we observed significant changes in the gene expression of cardiac markers of overload and remodeling in UUO males treated with Sigmar1 agonists for 21 days, we decided to investigate the transcriptomic landscape in the hearts of these mice treated with PRE-084 using RNA-seq. RNA-seq identified 101 DEGs with a log2 (fold-change) > 1 (Appendix A). A heatmap analysis illustrates DEG’s expression pattern differences between experimental groups and replicates (Figure 7A). Among these, eleven DEGs, five upregulated and six downregulated, were significantly and differentially expressed (*p* < 0.05) between PRE-084 and control UUO hearts (Figure 7B). A volcano map reflects the distribution of DEGs (Figure 7C). Among these DEGs stand out Iqcj and Schip1 fusion protein (Iqschfp), solute carrier family 6 (neurotransmitter transporter) member 20B (Slc6a20b), NADPH dependent diflavin oxidoreductase 1 (Ndor1), formyl peptide receptor 1 (Fpr1), and protocadherin gamma subfamily A 4 (Pcdhga4) as upregulated, and ribonucleoprotein PTB-binding 1 (Raver1), anoctamin 5 (Ano5), protocadherin gamma subfamily A 10 (Pcdhga10), activity-dependent neuroprotective protein (Adnp), septin 2 (Septin2), and nicotinamide nucleotide transhydrogenase (Nnt) as downregulated. The functions of many of these genes have been described as being in the central nervous system and heart, with predicted and validated functions in calcium-mediated responses, amino acid transport, pro-drug metabolism, inflammatory response, immune cell recruitment, G protein-coupled receptor signaling, muscle contraction, neural differentiation, actin polymerization, and redox state [26,27,28,29,30,31,32,33,34]. 

GO analysis of DEGs in UUO hearts of male mice treated with PRE-084 revealed significant enrichment in 10 GO terms within three independent categories: biological process (61), cellular component (84), and molecular function (45) (Appendix A). The 50 most enriched GO terms (*p* < 0.05) are shown in Figure 8A. Also, a scatter plot of the top 20 enriched GO terms indicates that upregulated genes are enriched in regulation of L-glutamate import across plasma membrane, positive regulation of polynucleotide adenylyltransferase activity, nitric oxide homeostasis, N-formyl receptor activity, negative regulation of cytoskeleton organization, scavenger receptor binding, complement receptor activity and cell signaling, and homophilic cell adhesion via plasma membrane adhesion molecules, among others (Figure 8B).

## 4. Discussion

Our study uncovered substantial findings on Sigmar1 in a CRS4 model induced by UUO in male and female mice. We found that obstructive nephropathy triggers renal and cardiac injury markers’ expression in a time- and sex-dependent manner, accompanied by elevated Sigmar1 in the kidneys and heart. Sigmar1 stimulation with its agonists PRE-084 and SA4503 worsens cardiac injury and fibrosis markers in both sexes, with a notably higher impact in males. 

The kidneys and heart are deeply interconnected organs, explaining why primary renal injury often leads to subsequent cardiac dysfunction [1,2,3]. Most pre-clinical CKD models use nephrectomy or bilateral injury to generate severe kidney damage and sustained uremia, emulating advanced stages of the disease [35,36,37]. However, in early-stage CKD patients, there is evidence of sub-clinical deterioration in cardiac performance, early signs of hypertrophy, and fibrosis without overt cardiovascular disease [38,39]. Previously, we showed widespread cardiac transcriptomic changes in a murine model of CRS4 induced by UUO after 21 days that resembled the mild uremia featured in early CKD [18,40,41,42]. This study used the same UUO model considering two earlier time points, seven and twelve days, and female mice. Although we observed only moderate increases in sCr and BUN after UUO surgery, a significant rise in renal Ngal expression occurred at all time points, but only in females. Since epithelial cells located in the loop of Henle express and excrete Ngal depending on the degree of tubular damage [43,44], our results indicate that female kidneys subjected to UUO suffer a more sustained injury than their male counterparts, also manifested as higher sCr levels at later times. These findings contradict ischemic murine models in which tubular damage, glomerular collapse, and leukocyte infiltration are more severe in males than females [16]. Notably, the degree of tubular injury observed in females did not translate into a higher expression of cardiac markers of dysfunction and remodeling than in males. Overall, our results coincide to some extent with CRS4 murine models reporting elevated markers of cardiac dysfunction, hypertrophy, and fibrosis, thus reinforcing the appropriateness of the UUO murine model to study early molecular changes associated with cardiac remodeling, without the burden of severe renal failure and uremia in both sexes [45,46]. Still, renal and cardiac injury develops differently in males and females, reinforcing the importance of reducing sex bias in pre-clinical studies.

Ham O et al. performed echocardiography to evaluate cardiac structure and function in UUO mice 21 days after the surgery and found an 18% increase in left ventricular mass in the UUO mice compared to controls, consistent with the pathological cardiac hypertrophy they observed in these animals, but without significant changes in cardiac function [46]. Their results suggest that sub-clinical cardiac dysfunction might be present in UUO hearts, but standard echocardiography cannot detect it. UUO mice also had significantly elevated systolic and diastolic blood pressures. Although these data indirectly support our study as we used the same early CKD model, there is the limitation that the authors only used male mice. Further studies are needed to evaluate how induced obstructive nephropathy leads to subsequent cardiac dysfunction and the role of sexual dimorphism. 

Sexual dimorphism and stress-activated signaling pathways are relevant to CRS4 development, thus, we investigated whether obstructive nephropathy altered the expression of the multifunctional chaperone Sigmar1. UUO induced Sigmar1 expression in the kidneys of male and female mice, and its levels augmented as the damage progressed. We found Sigmar1 predominantly localized in tubular structures within renal tissues and, unlike formerly described by Hosszu A. et al., in a few glomeruli [16,25]. Sigmar1 induction has been documented in other models of renal pathology [12]. Sigmar1 expression changes in the kidneys of diabetic male rats, increasing after two months of disease induction in distal tubules co-expressing TGF-β [47]. Meanwhile, Sigmar1 expression augmented in an IR rat model as early as two hours after reperfusion was higher in females than in males and ovariectomized rats, and correlated with improved renal recovery and survival [16]. 

Importantly, our study is the first to evaluate changes in Sigmar1 cardiac expression due to a primary kidney injury. We found that cardiac tissues express high levels of Sigmar1, regardless of treatment, further supporting Sigmar1’s physiological functions in regulating mitochondrial organization, remodeling, contractility, and organ size [48]. Yet, obstructive nephropathy increased Sigmar1 gene expression at day 12 post-surgery in male and female hearts. Our results contrast with findings in human hearts of methamphetamine users that develop cardiomyopathy in which Sigmar1 levels are significantly decreased [49] and in chronic heart failure models that found a reduction in Sigmar1 cardiac expression in rats with left anterior descending artery ligation [24]. Our study thus shows that UUO induces Sigmar1 expression in both kidneys and hearts, which might have important consequences for the pharmacological modulation of CRS4.

Elevated Sigmar1 in the heart of UUO mice could have a functional aspect associated with its role as a cytoprotective inter-organelle signaling molecule [6]. Multiple studies have investigated Sigmar1 regulation of cardiac physiology, pathology, and hemodynamics using Sigmar1 knockout mice. Sigmar1^−/−^ hearts have increased fibrosis, develop cardiac contractile dysfunction as they age, and present altered mitochondrial dynamics [48]. In Takotsubo syndrome-like cardiomyopathy induced by isoproterenol, Sigmar1 knockout mice exhibit aggravated cardiac dysfunction, ventricular remodeling, and gut microbiota dysbiosis [50,51]. This evidence leads us to speculate that increased Sigmar1 expression is a stress response to counteract contractile dysfunction and fibrosis resulting from renal injury. Additional experiments in Sigmar1 knockout mice would be essential to determine how Sigmar1 stress response becomes maladaptive during CRS4 development.

Multiple pre-clinical studies have suggested generating novel protective therapeutic options for renal and cardiovascular diseases through Sigmar1 activation. The Sigmar1 agonists dehydroepiandrosterone and fluvoxamine protect against renal IR injury in male rats by improving survival and renal function while reducing inflammation [25]. Moreover, estradiol-dependent Sigmar1 activation prevents renal IR injury by enhancing the heat-shock response [16]. There is also evidence of Sigmar1 agonist-mediated antifibrotic responses in trabecular meshwork cells and in a rat model of adenine-induced CKD, in which PRE-084 decreased sCr, proteinuria, and extra-cellular matrix deposition [52,53]. Similarly, PRE-084 and SA4503 administration in our study partially improved renal function by decreasing sCr and BUN levels, especially in males, but failed to reduce Ngal expression. Furthermore, Ngal was over-expressed at least two-fold in male kidneys, suggesting Sigmar1 modulation with its agonists PRE-084 and SA4503, and antagonist haloperidol, promotes tubular damage after UUO. 

Anp and Bnp maintain cardiovascular homeostasis by regulating vascular tone, intravascular volume, metabolism, and neurohormonal activity [20]. These biologically active natriuretic peptides also participate in CRS4 pathophysiology, and their synthesis and secretion are elevated under pathological heart conditions, making them legitimate markers of acute and chronic heart failure [54]. Various studies have indicated that Anp and Bnp have anti-fibrotic effects by inhibiting cell proliferation and extracellular matrix production induced by TGF-β in cardiac fibroblasts and Angiotensin II-dependent myocardial fibrosis [55,56,57]. We found that PRE-084 and SA4503 administration enhances cardiac Anp and Bnp expression more than 100-fold in males after UUO, highlighting a possible role of testosterone in further promoting Anp and Bnp expression. Since we did not evaluate Anp and Bnp circulating levels, we cannot ascertain whether significant increases in mRNA expression translate to higher secretion and function of these peptides. However, it is well documented that elevated Anp and Bnp levels in heart failure correlate with disease severity and possess prognostic value [58]. Haloperidol’s impact on renal and cardiac markers of dysfunction did not oppose those of Sigmar1’s agonists, perhaps due to haloperidol not being a pure antagonist. Haloperidol is an anti-psychotic that acts as a D2, D3, and D4 dopamine receptor antagonist, and in renal tubules, dopamine and atrial natriuretic peptides act concertedly to promote sodium excretion [59,60]. Additionally, it has been suggested that there is a potential interaction between estrogen and the dopaminergic system, evidenced by estrogen increasing the efficacy of haloperidol as an anti-psychotic medication, which could also influence the differences observed in relation to sex [61]. Thus, we cannot exclude that the effects of haloperidol in our CRS4 model are due to its dopaminergic regulation.

Treatment with Sigmar1 agonists resulted in a considerable increase in Anp and Bnp expression only in male hearts. Still, we observed that Sigmar1 agonists elevated cardiac collagen deposition and Acta2 levels equally in both sexes, indicating that Sigmar1 exacerbates pro-fibrotic mechanisms in the heart, secondary to obstructive nephropathy. Strikingly, our results contradict multiple reports proposing Sigmar1 as a therapeutic target for cardiac fibrosis. In murine models, agonistic stimulation of Sigmar1 decreases cardiac fibroblast activation and fibrosis induced by pressure overload, heart failure, pulmonary arterial hypertension, and ventricular arrhythmias [62,63,64,65]. In these models, Sigmar1 expression decreased as the damage progressed, and agonistic stimulation recovered Sigmar1 expression to induce cardioprotection. Here, CRS4 increased Sigmar1 expression in the heart, and the agonist or antagonist stimulation failed to significantly influence Sigmar1 expression at the protein level. These differences could be clues to understanding why, in this CRS4 model, Sigmar1 promoted injury instead of providing protection. In this study, we found that mRNA and protein expression levels did not increase proportionally for various genes, including Sigmar1. The differences are probably dependent on post-transcriptional regulation, which includes RNA-binding proteins, alternative untranslated regions influencing protein synthesis, different protein degradation rates, etc. [66]. Yet, transcriptional, and post-transcriptional regulation shape the cardiac response elicited by renal dysfunction, and more in-depth studies are needed to fully understand their contribution to CRS4 development.

In contrast, the Sigmar1 antagonist haloperidol increased Acta2 levels in cardiac tissues while severely attenuating Acta2 gene expression in males but did not affect UUO-induced collagen deposition. Rehman M et al. found that haloperidol strongly prevents myofibroblast activation through Sigmar1 and modulates downstream intracellular calcium, ER stress response, Notch1 signaling, and its target Acta2 [67]. Moreover, chronic administration of haloperidol to mice for ten days after acute myocardial infarction resulted in a reduced scar size and Acta2-positive myofibroblasts. However, the early effect on scar size was not evident at later time points, like in our study, perhaps due to haloperidol’s adverse cardiac effects overcoming its ability to control myofibroblast activation via Acta2 downregulation. Others have also shown that haloperidol causes mitochondrial dysfunction, and this aggravates trans-verse aortic constriction-induced heart failure [68]. 

Sigmar1 signaling is very complex and context-dependent [6,12]. A limitation of our study is that we did not analyze downstream signaling pathways activated by Sigmar1 in our model to dissect better how Sigmar1 exerts its effects on cardiac remodeling induced by UUO, which is something to consider for future studies. However, we did examine the impact of Sigmar1 stimulation with PRE-084 on transcriptional regulation and enriched pathways in male hearts showing a more aggravated cardiac remodeling. Sigmar1 stimulation resulted in differential expression of genes (Slc6a20b, Ndor1, Fpr1, Pcdhga4, Raver1, Ano5, Pcdhga10, Adnp, Septin2, and Nnt) with predicted and validated functions in calcium-mediated responses, amino acid transport, pro-drug metabolism, inflammatory response, immune cell recruitment, G protein-coupled receptor signaling, muscle contraction, neural differentiation, actin polymerization, and redox state [26,27,28,29,30,31,32,33,34]. These genes could provide unique insights into CKD-related cardiac pathways that contribute to CRS4 progression. However, their direct contribution and how these changes differ in females need to be validated experimentally. 

## 5. Conclusions

This study demonstrates that obstructive nephropathy increases Sigmar1 expression in the kidneys and heart, and that chronically stimulating Sigmar1 worsens heart function and remodeling, with a more significant impact on males. These findings should be considered when contemplating Sigmar1 as a pharmacological target in CRS4.

## Figures and Tables

**Figure 1 biomedicines-12-01908-f001:**
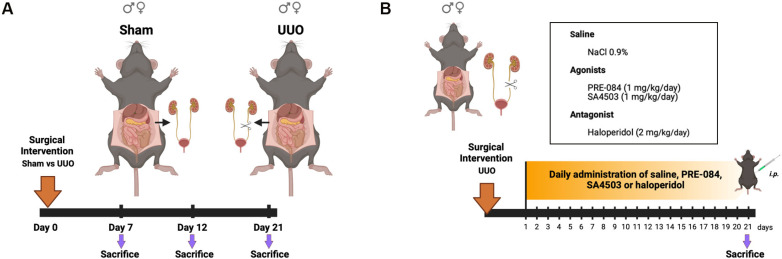
Experimental design. (**A**) Sham vs. UUO time course (7, 12, and 21 days) in male and female mice. (**B**) UUO male and female mice treated daily with saline, PRE-084 (1 mg/kg), SA4503 (1 mg/kg), or haloperidol (2 mg/kg), and sacrificed at day 21. Subsequent evaluation of serum, left kidney, and heart was carried out in all groups, N = 5.

**Figure 2 biomedicines-12-01908-f002:**
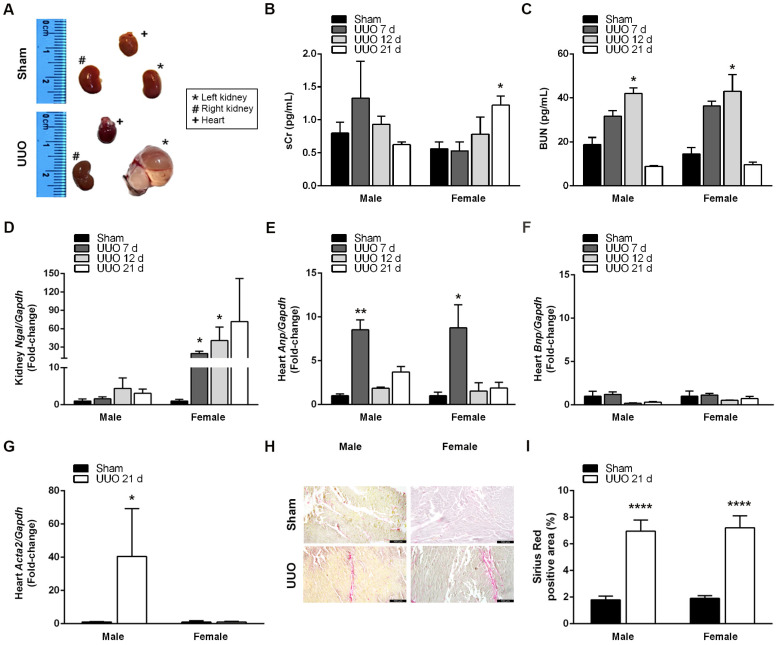
UUO induces CRS4 development. (**A**) Representative images of kidneys and hearts from sham and UUO male mice after 21 days. Circulating levels of (**B**) sCR (mg/dL) and (**C**) BUN (mg/dL) in male and female mice after sham surgery or UUO (7, 12, and 21 days). Fold-change of renal (**D**) Ngal and cardiac (**E**) Anp, (**F**) Bnp, and (**G**) Acta2 determined by RT-qPCR and normalized with Gapdh in male and female mice after sham surgery or UUO (7, 12, and 21 days). (**H**) Representative images of sham and UUO hearts after 21 days stained with Sirius Red (200×, scale bar = 100 μm). (**I**) Percentage of Sirius Red-positive area in sham and UUO hearts after 21 days. N = 5, * *p* < 0.05; ** *p* < 0.01; **** *p* < 0.0001 vs. sham.

**Figure 3 biomedicines-12-01908-f003:**
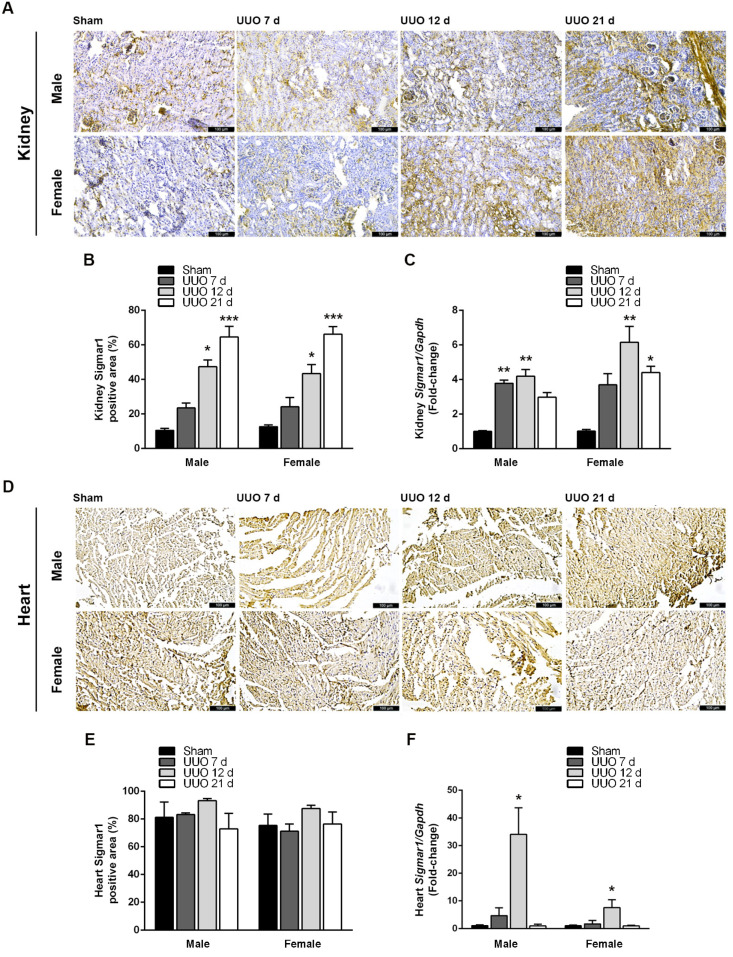
UUO elevates Sigmar1 expression in the kidney and heart. (**A**) Representative images of anti-Sigmar1 IHC kidney sections from sham and UUO (7, 12, and 21 days) mice (200×, scale bar = 100 μm). (**B**) Percentage of Sigmar1 positive area in sham and UUO (7, 12, and 21 days) kidneys. (**C**) Fold-change of renal Sigmar1/Gapdh by RT-qPCR in male and female mice after sham surgery or UUO (7, 12, and 21 days). (**D**) Representative images of anti-Sigmar1 IHC heart sections from sham and UUO (7, 12, and 21 days) mice (200×, scale bar = 100 μm). (**E**) Percentage of Sigmar1-positive area in sham and UUO (7, 12, and 21 days) hearts. (**F**) Fold-change of cardiac Sigmar1/Gapdh by RT-qPCR in male and female mice after sham surgery or UUO (7, 12, and 21 days). N = 5, * *p* < 0.005; ** *p* < 0.01, *** *p* < 0.001 vs. sham.

**Figure 4 biomedicines-12-01908-f004:**
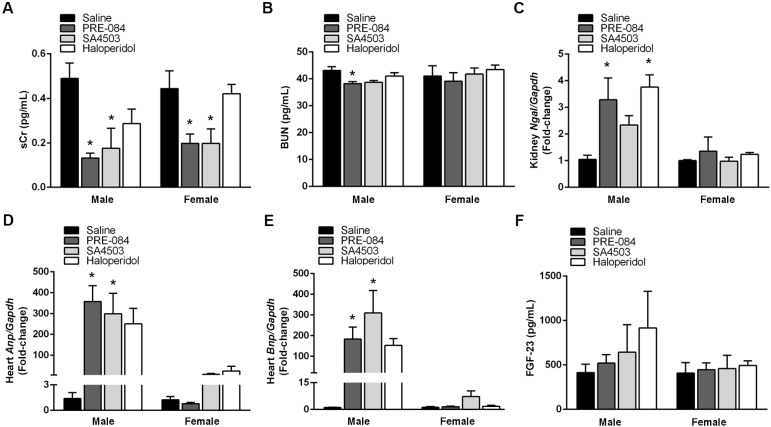
Male sex potentiates Sigmar1 effects on renal and cardiac function markers after UUO. Circulating levels of (**A**) sCR (mg/dL) and (**B**) BUN (mg/dL) in male and female mice after 21 days of UUO and treatment with saline, PRE-084 (1 mg/kg/day), SA4503 (1 mg/kg/day), and haloperidol (2 mg/kg/day). Fold-change of renal (**C**) Ngal and cardiac (**D**) Anp, and (**E**) Bnp determined by RT-qPCR and normalized with Gapdh in male and female mice after 21 days of UUO and treatment with saline, PRE-084, SA4503, and haloperidol. (**F**) Renal concentration of FGF-23 (pg/mL) per μg of protein quantified by ELISA in male and female mice after 21 days of UUO and treatment with saline, PRE-084, SA4503, and haloperidol. N = 5, * *p* < 0.005 vs. saline.

**Figure 5 biomedicines-12-01908-f005:**
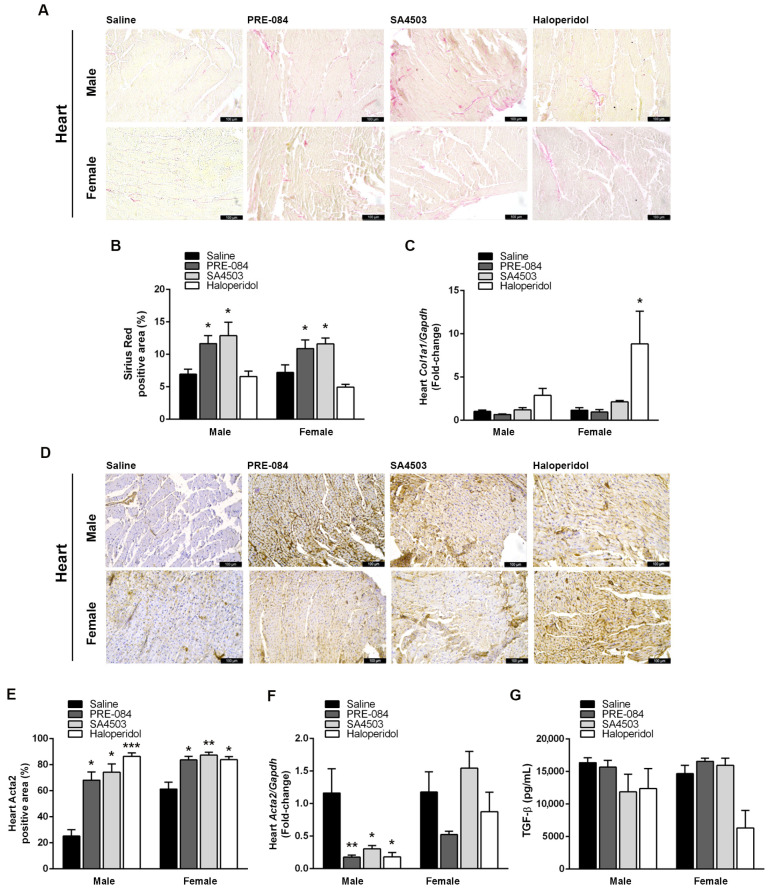
Sigmar1 agonists and antagonist potentiate cardiac fibrosis development after UUO. (**A**) Representative images of cardiac sections stained with Sirius Red (200×, scale bar = 100 μm) after 21 days of UUO and treatment with saline, PRE-084 (1 mg/kg/day), SA4503 (1 mg/kg/day), and haloperidol (2 mg/kg/day). (**B**) Percentage of Sirius Red-positive area in male and female hearts after 21 days of UUO and treatment with saline, PRE-084, SA4503, and haloperidol. (**C**) Cardiac Col1a1/Gapdh fold-change by RT-qPCR in male and female mice after 21 days of UUO and treatment with saline, PRE-084, SA4503, and haloperidol. (**D**) Representative images of anti-Acta2 IHC cardiac sections after 21 days of UUO and treatment with saline, PRE-084, SA4503, and haloperidol (200×, scale bar = 100 μm). (**E**) Percentage of Acta2-positive area in male and female hearts after 21 days of UUO and treatment with saline, PRE-084, SA4503, and haloperidol. (**F**) Cardiac Acta2/Gapdh fold-change by RT-qPCR in male and female mice after 21 days of UUO and treatment with saline, PRE-084, SA4503, and haloperidol. (**G**) Serum TGF-β (pg/mL) quantified by ELISA in male and female mice after 21 days of UUO and treatment with saline, PRE-084, SA4503, and haloperidol. N = 5, * *p* < 0.005; ** *p* < 0.01; *** *p* < 0.001 vs. saline.

**Figure 6 biomedicines-12-01908-f006:**
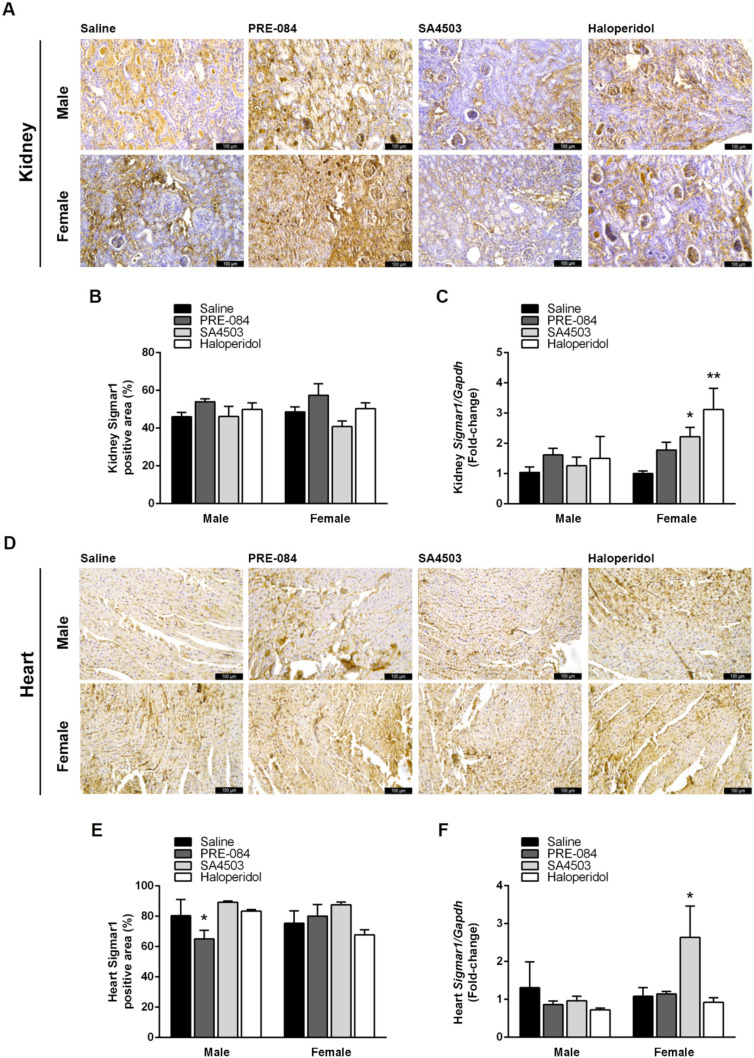
Sigmar1 expression in the kidneys and heart is modulated by its agonists and antagonists after UUO. (**A**) Representative images of anti-Sigmar1 IHC kidney sections after 21 days of UUO and treatment with saline, PRE-084 (1 mg/kg/day), SA4503 (1 mg/kg/day), and haloperidol (2 mg/kg/day) (200×, scale bar = 100 μm). (**B**) Percentage of Sigmar1-positive areas in kidneys of males and females after 21 days of UUO and treatment with saline, PRE-084, SA4503, and haloperidol. (**C**) Renal Sigmar1/Gapdh fold-change by RT-qPCR after 21 days of UUO and treatment with saline, PRE-084, SA4503, and haloperidol. (**D**) Representative images of anti-Sigmar1 IHC cardiac sections after 21 days of UUO and treatment with Saline, PRE-084, SA4503, and haloperidol (200×, scale bar = 100 μm). (**E**) Percentage of Sigmar1-positive areas in male and female hearts after 21 days of UUO and treatment with saline, PRE-084, SA4503, and haloperidol. (**F**) Cardiac Sigmar1/Gapdh fold-change by RT-qPCR in male and female mice after 21 days of UUO and treatment with saline, PRE-084, SA4503, and haloperidol. N = 5, * *p* < 0.005; ** *p* < 0.01 vs. saline.

**Figure 7 biomedicines-12-01908-f007:**
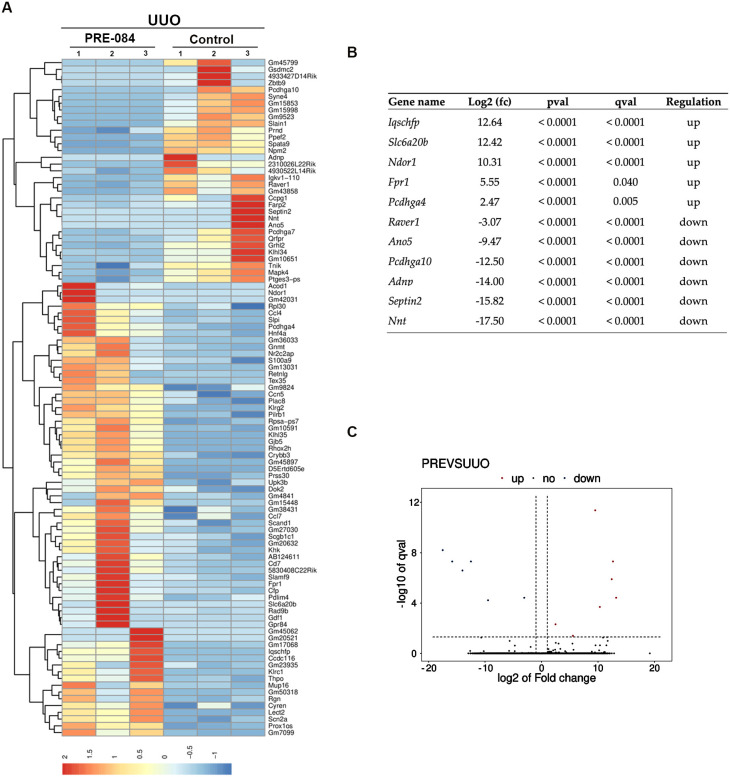
Cardiac DEGs induced by PRE-084 in UUO male mice. (**A**) Heatmap plot of DEGs in PRE-084-treated male mice for 21 days after UUO surgery. Red = upregulated and blue = downregulated. *p* < 0.05 and log2 (fold change) > 1. (**B**) List of most significantly enriched DEGs. (**C**) Volcano map showing the distribution of cardiac DEGs in PRE-084-treated male mice for 21 days after UUO surgery. N = 3.

**Figure 8 biomedicines-12-01908-f008:**
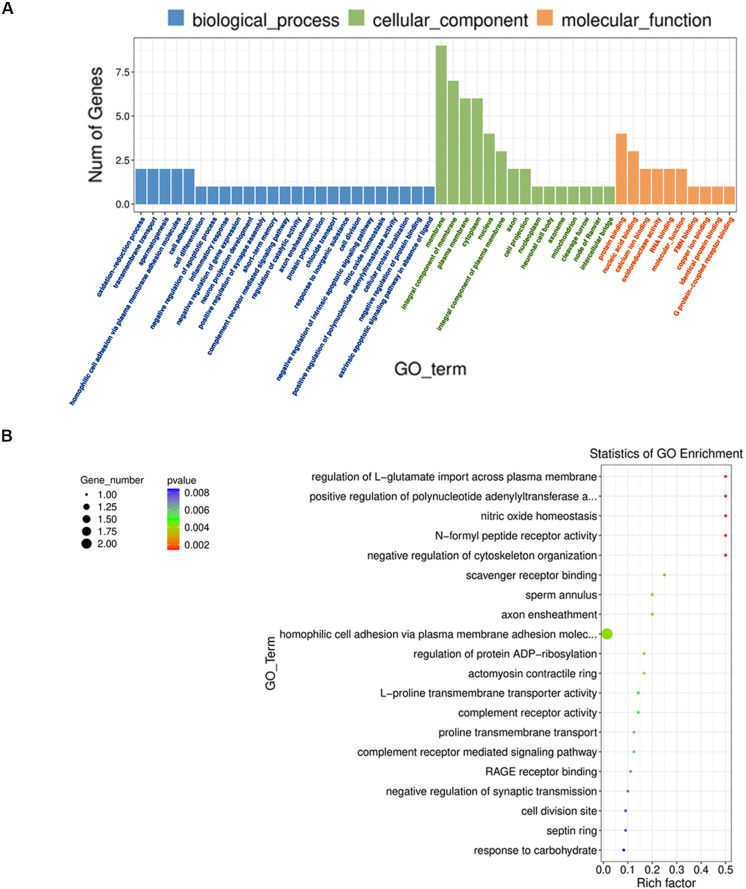
Enrichment analysis in UUO hearts treated with PRE-084. (**A**) Enriched GO terms. (**B**) Scatter plot of the top enriched GO terms. Circle sizes, number of enriched genes per pathway, color, and *p*-value range. N = 3.

## Data Availability

The original contributions presented in the study are openly available in the NCBI GEO repository under accession code GSE235751 and are included in the article/Appendix A. Further inquiries can be directed to the corresponding author.

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
