# Peer review of "The Sigma-1 Receptor Exacerbates Cardiac Dysfunction Induced by Obstructive Nephropathy: A Role for Sexual Dimorphism"

_biomedicines, 2024, doi:10.3390/biomedicines12081908_

Round 1

Reviewer 1 Report

Comments and Suggestions for Authors

In this study, the authors investigated the role of the stress response Sigmar1 receptor in cardiorenal syndrome type 4 (CRS4) induced by unilateral ureteral obstructions (UUO) in male and female mice. They evaluated markers of renal and cardiac dysfunction, Sigmar1 expression, and cardiac remodeling over time (7, 12, and 21 days) after the chronic administration of the Sigmar1 agonists PRE-084 and SA4503, as well as the antagonist haloperidol for 21 days. The authors found that obstructive nephropathy triggers the expression of renal and cardiac injury markers in a time- and sex-dependent manner, accompanied by elevated Sigmar1 levels in both the kidney and heart. Stimulation of Sigmar1 with its agonists PRE-084 and SA4503 worsens cardiac injury and fibrosis markers in both sexes, with a notably greater impact observed in males. The described approach is very promising, and the results are of significant interest. However, I have a few questions for the authors.

Major

1. The author referred that the mouse induced obstructive nephropathy leads subsequently to cardiac dysfunction resulting biomarker measured cardiac fibrosis. This is really interesting and it would strength the results if the authors could provide some functional heart measurements of their UUO model like echocardiography to point out time and sex dependent manner.

2.      The elevated expression of sigmar1 in the heart of the UUO Model in the 12 days group seems to have a functional aspect. Is there any explanation from the authors how this could be explained? Is the stress response a reason for that? How the authors think about using a sigmar1 knock out model to test their hypothesis (DOI:10.3390/biomedicines11102766 Simgar 1 receptor and cardiac injury KO Mouse model).

Minor

1.      Figure 2 H/I – The images do not match the quantification. In fact, the UUO group should exhibit a stronger red staining than the sham group. This is not visible in the histological sections. Rather, it appears that the UUO group shows a lower proportion of red staining. Additionally, the heart muscle is not optimally visible in the sections, especially in the sham images.

2.      Figure 3 Sigmar1 expression. It looks like the SHAM operation alone already has an effect on the Sigmar1 expression in the heart. Are there any prior data on the expression of Sigmar1 in the hearts of untreated wild-type mice? What are the reasons for the differences in expression at the protein and gene levels?

3.      Please check the label of Figure 5; the assignment to the legend is difficult because numbers are missing in the illustrations.

Author Response

Response to Reviewer 1

Dear reviewer, we appreciate your time and comments to improve our manuscript. We modified the text and figures in response to your observations. All the changes are visible in the manuscript (red). You can find a detailed response to each of your concerns below.

In this study, the authors investigated the role of the stress response Sigmar1 receptor in cardiorenal syndrome type 4 (CRS4) induced by unilateral ureteral obstructions (UUO) in male and female mice. They evaluated markers of renal and cardiac dysfunction, Sigmar1 expression, and cardiac remodeling over time (7, 12, and 21 days) after the chronic administration of the Sigmar1 agonists PRE-084 and SA4503, as well as the antagonist haloperidol for 21 days. The authors found that obstructive nephropathy triggers the expression of renal and cardiac injury markers in a time- and sex-dependent manner, accompanied by elevated Sigmar1 levels in both the kidney and heart. Stimulation of Sigmar1 with its agonists PRE-084 and SA4503 worsens cardiac injury and fibrosis markers in both sexes, with a notably greater impact observed in males. The described approach is very promising, and the results are of significant interest. However, I have a few questions for the authors.

Major

  1. The author referred that the mouse induced obstructive nephropathy leads subsequently to cardiac dysfunction resulting biomarker measured cardiac fibrosis. This is really interesting and it would strength the results if the authors could provide some functional heart measurements of their UUO model like echocardiography to point out time and sex dependent manner. 

Response: We appreciate your suggestion and agree that hemodynamic and echocardiographic data would strengthen our results, but unfortunately, we could not perform these experiments. However, Ham O et al. performed echocardiography to evaluate cardiac structure and function in UUO mice 21 days after the surgery and found an 18% increase in Left ventricular mass in the UUO mice compared to controls, consistent with the pathological cardiac hypertrophy they observed in these animals, but without significant changes in cardiac function [46]. Their results suggest that subclinical cardiac dysfunction might be present in UUO hearts, but standard echocardiography cannot detect it. UUO mice also had systolic and diastolic blood pressures significantly elevated. Although this data indirectly supports our study as we used the same early CKD model, there is the limitation that the authors only used male mice. Further studies are needed to evaluate how induced obstructive nephropathy leads to subsequent cardiac dysfunction and the role of sexual dimorphism. (Added in Lines 426-435).

  1. The elevated expression of sigmar1 in the heart of the UUO Model in the 12 days group seems to have a functional aspect. Is there any explanation from the authors how this could be explained? Is the stress response a reason for that? How the authors think about using a sigmar1 knock out model to test their hypothesis (DOI:3390/biomedicines11102766Sigmar 1 receptor and cardiac injury KO Mouse model). 

Response: Elevated Sigmar1 in the heart of UUO mice could have a functional aspect associated with its role as a cytoprotective interorganelle signaling molecule [6]. Multiple studies have investigated Sigmar1 regulation of cardiac physiology, pathology, and hemodynamics using Sigmar1 knockout mice. Sigmar1−/− hearts have increased fibrosis, develop cardiac contractile dysfunction as they age, and present altered mitochondrial dynamics [48]. In Takotsubo syndrome-like cardiomyopathy induced by isoproterenol, Sigmar1 knockout mice exhibit aggravated cardiac dysfunction, ventricular remodeling, and gut microbiota dysbiosis [50,51]. This evidence leads us to speculate that increased Sigmar1 expression is a stress response to counteract contractile dysfunction and fibrosis resulting from renal injury. Additional experiments in Sigmar1 knockout mice would be essential to determine how Sigmar1 stress response becomes maladaptive during CRS4 development. (Added in Lines 459-469).

Minor 

  1. Figure 2 H/I – The images do not match the quantification. In fact, the UUO group should exhibit a stronger red staining than the sham group. This is not visible in the histological sections. Rather, it appears that the UUO group shows a lower proportion of red staining. Additionally, the heart muscle is not optimally visible in the sections, especially in the sham images.

Response: We thank you for the observation. The images have been updated to better represent the results from the quantification.

  1. Figure 3 Sigmar1 expression. It looks like the SHAM operation alone already has an effect on the Sigmar1 expression in the heart. Are there any prior data on the expression of Sigmar1 in the hearts of untreated wild-type mice? What are the reasons for the differences in expression at the protein and gene levels?

Response: This is an interesting point. We did not determine basal Sigmar1 expression in the hearts of mice that had not undergone surgery. In the literature, we did not find specific data on Sigmar1 expression in wild-type mice. However, in the Human Protein Atlas (proteinatlas.org), the human heart shows a low-moderate basal Sigmar1 expression that varies according to cell type (cardiomyocytes<endothelial cells<fibroblasts). In this study, we found that mRNA and protein expression levels did not increase proportionally for various genes including Sigmar1. The differences are probably dependent on post-transcriptional regulation, which includes RNA-binding proteins, alternative untranslated regions influencing protein synthesis, different protein degradation rates, etc. [66]. Yet, transcriptional, and post-transcriptional regulation shape the cardiac response elicited by renal dysfunction, and more in-depth studies are needed to fully understand their contribution to CRS4 development. (Added in Lines 518-525).

  1. Please check the label of Figure 5; the assignment to the legend is difficult because numbers are missing in the illustrations.

Response: We appreciate the observation and corrected Figure 5 to include the missing data.

Reviewer 2 Report

Comments and Suggestions for Authors

In this manuscript, the authors demonstrate that obstructive nephropathy induces Sigmar1 expression in both the kidney and heart and that stimulation of Sigmar1 with agonists exacerbates cardiac dysfunction and remodeling in both sexes, with more pronounced effects in males. These findings underscore important sex-related differences in the development of CRS4, which should be considered when evaluating Sigmar1 as a potential pharmacological target. Sigmar1 is a stress-activated chaperone and a promising target for pharmacological intervention due to its role in triggering various cellular responses. The results are convincing and well-explained, but I have a few suggestions:

1. It would be helpful to explain the rationale for assessing markers of cardiac hypertrophy (Anp1/Bnp1) and renal function in the methods section rather than introducing them directly in the results. Additionally, specify which part of the heart tissue was used for IHC/qPCR analyses.

2. In Figure 4A, please recheck the significance level for females. In Figure 4C, it’s unclear why NGAL expression is elevated in females treated with the Sigmar1 antagonist—please clarify.

3. Figure 5 lacks significance levels, and there is no sub-labeling (e.g., A, B, C, etc.) as indicated in the legend. Additionally, consider measuring protein expression (WB) for ACTA2.

4. In Figure 6, the authors state that Sigmar1 expression in the kidney and heart is modulated by its agonists and antagonists after UUO. However, figure 6B shows no significant difference between saline and agonist or antagonist treatments in males and females, while RNA expression is higher in the antagonist-treated group in females. This is confusing—please review and clarify.

5. Why were protein levels for Anp1/Bnp1 not measured? If tissue samples are available, consider performing a western blot.

6. Yang JZ et al. reported that Sigma-1 receptor knockout disrupts gut microbiota, alters the serum metabolome, and worsens isoprenaline-induced heart failure (PMCID: PMC10501138). It is recommended to discuss this paper in your manuscript.

7. Please watch for typos, such as "im-pact" in line 400

Author Response

Response to Reviewer 2

Dear reviewer, we appreciate your time and comments to improve our manuscript. We modified the text and figures in response to your observations. All the changes are visible in the manuscript (red). You can find a detailed response to each of your concerns below.

In this manuscript, the authors demonstrate that obstructive nephropathy induces Sigmar1 expression in both the kidney and heart and that stimulation of Sigmar1 with agonists exacerbates cardiac dysfunction and remodeling in both sexes, with more pronounced effects in males. These findings underscore important sex-related differences in the development of CRS4, which should be considered when evaluating Sigmar1 as a potential pharmacological target. Sigmar1 is a stress-activated chaperone and a promising target for pharmacological intervention due to its role in triggering various cellular responses. The results are convincing and well-explained, but I have a few suggestions:

  1. It would be helpful to explain the rationale for assessing markers of cardiac hypertrophy (Anp1/Bnp1) and renal function in the methods section rather than introducing them directly in the results. Additionally, specify which part of the heart tissue was used for IHC/qPCR analyses.

Response: The information was added to the methods section (Line 121)

  1. In Figure 4A, please recheck the significance level for females. In Figure 4C, it’s unclear why NGAL expression is elevated in females treated with the Sigmar1 antagonist—please clarify.

Response: We have updated Figure 4A. In Figure 4C, Ngal expression was found to be elevated only in males, not in females. However, the effect seen with the antagonist haloperidol could be due to its known ability to also antagonize dopamine receptors. The following explanation is given in Lines 496-505: Haloperidol's impact on renal and cardiac markers of dysfunction did not oppose those of Sigmar1 's agonists, perhaps due to haloperidol not being a pure antagonist. Haloperidol is an antipsychotic that acts as a D2, D3, and D4 dopamine receptor antagonist, and in renal tubules, dopamine, and atrial natriuretic peptides act concertedly to promote sodium excretion [59,60]. Additionally, it has been suggested that there is a potential interaction between estrogen and the dopaminergic system, evidenced by estrogen in-creasing the efficacy of haloperidol as an antipsychotic medication, which could also influence the differences observed in relation to sex [61]. Thus, we cannot exclude that the effects of haloperidol in our CRS4 model are due to its dopaminergic regulation.

  1. Figure 5 lacks significance levels, and there is no sub-labeling (e.g., A, B, C, etc.) as indicated in the legend. Additionally, consider measuring protein expression (WB) for ACTA2.

Response: We appreciate the observation and updated Figure 5 to include the missing data. As of measuring Acta2 protein expression with western blot, unfortunately we could not perform these experiments.

  1. In Figure 6, the authors state that Sigmar1 expression in the kidney and heart is modulated by its agonists and antagonists after UUO. However, figure 6B shows no significant difference between saline and agonist or antagonist treatments in males and females, while RNA expression is higher in the antagonist-treated group in females. This is confusing—please review and clarify.

 Response: In this study, we found that mRNA and protein expression levels did not increase proportionally for various genes including Sigmar1. The differences are probably dependent on post-transcriptional regulation, which includes RNA-binding proteins, alternative untranslated regions influencing protein synthesis, different protein degradation rates, etc. [66]. Yet, transcriptional, and post-transcriptional regulation shape the cardiac response elicited by renal dysfunction, and more in-depth studies are needed to fully understand their contribution to CRS4 development. (Added in Lines 518-525)

  1. Why were protein levels for Anp1/Bnp1 not measured? If tissue samples are available, consider performing a western blot.

Response: This is a good suggestion. Unfortunately, we do not have tissue samples available to determine Anp/Bnp protein levels with western blot.

  1. Yang JZ et al. reported that Sigma-1 receptor knockout disrupts gut microbiota, alters the serum metabolome, and worsens isoprenaline-induced heart failure (PMCID: PMC10501138). It is recommended to discuss this paper in your manuscript.

Response: Elevated Sigmar1 in the heart of UUO mice could have a functional aspect associated with its role as a cytoprotective interorganelle signaling molecule [6]. Multiple studies have investigated Sigmar1 regulation of cardiac physiology, pathology, and hemodynamics using Sigmar1 knockout mice. Sigmar1−/− hearts have increased fibrosis, develop cardiac contractile dysfunction as they age, and present altered mitochondrial dynamics [48]. In Takotsubo syndrome-like cardiomyopathy induced by isoproterenol, Sigmar1 knockout mice exhibit aggravated cardiac dysfunction, ventricular remodeling, and gut microbiota dysbiosis [50,51]. This evidence leads us to speculate that increased Sigmar1 expression is a stress response to counteract contractile dysfunction and fibrosis resulting from renal injury. Additional experiments in Sigmar1 knockout mice would be essential to determine how Sigmar1 stress response becomes maladaptive during CRS4 development. (Added in Lines 459-469).

  1. Please watch for typos, such as "im-pact" in line 400

Response: We revised the manuscript and corrected the typos.